# Fluid and solute transport by cells and a model of systemic circulation

**Yufei Wu[1,2], Morgan A. Benson[1,2], Sean X. Sun** (iD)[1,2,3]*

**1** Department of Mechanical Engineering, Johns Hopkins University, Baltimore, Maryland, United States of America, **2** Institute for NanoBioTechnology, Johns Hopkins University, Baltimore, Maryland, United States of America, **3** Center for Cell Dynamics, Johns Hopkins School of Medicine, Baltimore, Maryland, United States of America

\* ssun@jhu.edu

**Data availability statement:** All relevant data and models are included within the

## Abstract

Active fluid circulation and solute transport are essential functions of living organisms, enabling the efficient delivery of oxygen and nutrients to various physiological compartments. Since fluid circulation occurs in a network, the systemic flux and pressure are not simple outcomes of individual components. Rather, they are emergent properties of network elements and network topology. Moreover, consistent pressure and osmolarity gradients are maintained across compartments such as the kidney, interstitium, and blood vessels. The mechanisms by which these gradients and network properties are established and maintained are unanswered questions in systems physiology. Previous studies have shown that epithelial cells are fluid pumps and can actively generate pressure and osmolarity gradients. The polarization and activity of solute transporters in epithelial cells, which drive fluid flux, are influenced by pressure and osmolarity gradients. Therefore, there is an unexplored coupling between pressure and osmolarity in the circulatory network. In this work, we develop a mathematical framework that integrates the influence of pressure and osmolarity on solute transport. We use this model to explore both cellular fluid transport and systemic circulation. Using a simple network featuring the kidney-vascular interface, we show that our model naturally generates pressure and osmolarity gradients across the kidney, vessels and renal interstitium. While the current model uses this interface as an example, the findings can be generalized to other physiological compartments. This model demonstrates how systemic transport properties can depend on cellular properties and, conversely, how cell states are influenced by systemic properties. When epithelial and endothelial pumps are considered together, we predict how pressures at various points in the network depend on the overall osmolarity of the system. The model can be improved by including physiological geometries and expanding solute species, and highlights the interplay of fluid properties with cell function in living organisms.

manuscript and its Supporting information files. The code for solving the model is available on GitHub: https: //github.com/sxslabjhu/Systemic-circulation.git.

**Funding:** This work has been funded in part by the National Institutes of Health grants (R01GM134542 to SXS). The funders had no role in study design, data collection and analysis, decision to publish, or preparation of the manuscript.

**Competing interests:** The authors have declared that no competing interests exist.

## Author summary

Active fluid circulation and solute transport in living organisms are essential for delivering oxygen and nutrients to tissues. These processes rely on fluid circulatory networks, such as blood vessels, the kidney, and other organs. A defining feature of circulatory networks is the consistent presence of pressure and osmolarity (solute concentration) gradients across compartments. How these gradients are established and maintained remains unclear. Here we develop a mathematical model that links cellular-scale solute and fluid transport to systemic circulation. Using the kidney-vascular interface as an example, our model recapitulates the pressure and osmolarity gradients in these physiological systems and uncovers how network-wide circulation properties depend on cell solute transport activity. This framework highlights the intricate connections between cells and systemic behavior in circulation networks. When compartment boundaries are not static but are dynamic and growing, we can generalize the model to describe morphogenesis.

## Introduction

Fluid circulation across various physiological compartments is driven by overall blood flow and active pumping of solutes across epithelial and endothelial barriers. The overall circulation in humans is large (100s of liters per day), and can be considered essentially as a closed circuit. For example, in the kidney, fluid transport from the apical (lumen-facing) to the basal (interstitial-facing) side is primarily facilitated by the active transport of $Na^+$ and $Cl^-$ across the epithelial layer by ion transporters [1,2] (Fig 1). Since water follows solute transport, small solute concentration gradients will drive water flow and generate hydraulic pressure gradients. On the other hand, it was shown that the presence of hydraulic pressure gradients can influence cell ion channel apical-basal polarization and change solute/water flux [3]. Thus, pressure gradients can directly influence solute flux at the cellular level. Experimental data on the pressure dependence of solute and water transport has been available in the literature for epithelial cells [4–7], endothelial cells [8] and possibly others. Pressure natriuresis is also a well-known phenomena driven by blood pressure-dependence of water transport in the kidney [9,10]. Previous studies have explored the combined influence of pressure and osmolarity on water transport in single-cells and tissues. Various models have been developed for epithelial cells with active solute pumping [3,11–14]. However, the complex interplay between solute concentration, oncotic pressure, interstitial and blood hydraulic pressures in systemic circulation have not been considered theoretically. Moreover, pressure and osmolarity differences in the interstitium, vessels and capillaries, the lymphatic space, and subcutaneous compartments are known [15–18], which must be emergent properties of the overall circulation network. These facts suggest that a unified mathematical framework that starts with cell properties and ends with systemic fluid circulation predictions is needed.

In this paper, we develop a mathematical model of systemic fluid circulation that incorporates the active pumping characteristics of epithelial and endothelial cells within a network context. The model starts with cellular level transport properties, and incorporates pressure and osmolarity dependent solute transport at the single cell level. We then use the cell scale model to derive transport/pumping properties of epithelial and endothelial barriers. The results directly relate phenomenological coefficients of transport equations with cell-level properties. We then incorporate active pumping properties of cells in a circulation network model, accounting for both solute and fluid pumping in the network. This model predicts the

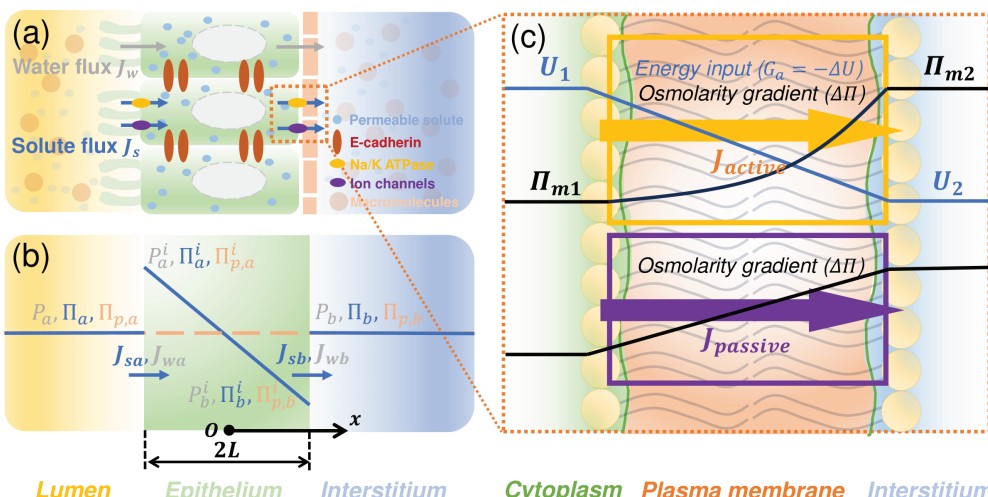

**Fig 1. Schematic illustration of the active water and solute transport model at the cellular level.** (a) An illustration of active water transport in epithelial layer. Water transport is driven by gradients of hydraulic pressure and osmotic pressure across the apical and basal cell membrane. The osmotic pressure arises from impermeable macromolecules (e.g., proteins) and permeable molecules (e.g., NaCl). The osmotic pressure gradient of permeable solute is established by both active ion pumping (e.g., Na/K ATPase) and passive ion transport (ion channels). (b) A diagram of the model and predicted spatial distribution of hydraulic pressure ($P$), osmotic pressure of permeable molecules ($\Pi$) and impermeable macromolecules ($\Pi_p$). Subscripts $a,b$ denote external apical and basal side of the epithelium, $^i_a, ^i_b$ denote apical and basal side of the cell interior. $J_w$ and $J_s$ denote water and solute flux, respectively. (c) Permeable solute flux across the membrane results from active pumping and passive diffusion. Active pumping is driven by a free energy function $U$ across the cell membrane. Passive solute flux is driven by the solute concentration gradient. $\Pi_m$ is the predicted solute concentration profile within the membrane (S1 Text).

overall flux of solutes and fluids across the network, as well as the osmolarity and pressure distributions within its various compartments. Gradients of osmolarity and pressure are natural outcomes of the model, and can influence each other. We show that inclusion of active transport or pumping properties of endothelia/epithelia fundamentally changes the overall network circulation properties. Specifically, the model predicts how the total osmolarities of the system can influence network transport properties and pressure/osmolarity gradients across compartments. We also point out how the model can be extended to include realistic network geometries and physiology-level feedback control. Mechanical rigidity of the network can be relaxed and growth can be included to allow the network morphology to adapt to pressure/osmolarity changes, leading to a fluid-centric theory of morphogenesis.

## Cell monolayer fluid and solute pumping model

The first element is to develop a model of a single epithelial/endothelial layer where fluid and solute transport are treated together. Details of this model are given in the supplemental material (S1 Text). The model considers a single neutral solute (e.g., NaCl) driven by active solute transporters. This is a simplification that does not include complexities of multiple ionic species and their possible interaction with electrical fields. A more detailed model with chemical complexity can be included in a follow-up model. In addition to small permeable solutes, we also include impermeable macromolecules such as proteins, which do not move through the membrane. The osmotic pressure of these macromolecules is denoted by $\Pi_p$. The cell layer is modeled as a domain of thickness $2L$ (Fig 1), where solute and water fluxes, $J_s$ and $J_w$, can occur at the two surfaces facing the lumen and the interstitium, which have osmotic

pressures and hydraulic pressures $(\Pi_a, \Pi_b, \Pi_{p,a}, \Pi_{p,b}, P_a, P_b)$, respectively. The primary principle governing these flows is the continuity of fluxes. In other words, $J_s$ and $J_w$ across the surfaces are continuous with respect to the fluxes in the cytoplasmic domain. The permeable solute flux in the cytoplasmic domain is a combination of diffusion plus convection:

$$J_s = -D\nabla\Pi + J_w\Pi. \tag{1}$$

where $\Pi = RTc$ is the osmotic pressure of the permeable solute, $c$ is the solute concentration. $D$ is the solute diffusion coefficient and $RT$ is the gas constant times temperature. For convenience, we incorporate the factor of $RT$ into solute flux $J_s$. The water flux in the cytoplasm, $J_w$, contributes to the convection of the solute. It is related to the hydraulic pressure gradient in the cytoplasm: $J_w \propto \nabla P$ (see S1 Text). Therefore the solute and water flux must be solved together in a coupled manner. In 1D, water flux (or velocity) in the cytoplasm becomes: $J_w \propto (P_a^i - P_b^i)/\mu L$. Here, $\mu$ is the dynamic viscosity of fluid. $P_{a,b}^i$ is the hydraulic pressure just inside the cell at the apical/basal surface.

To describe the 'pumping' behavior of solutes driven by active ion transporters and passive permeation, a model of permeable solute flux across the cell membrane is needed (Fig 1c). In general, near equilibrium, the flux of material is proportional to the free energy gradient and material concentration $c$: $J_s \propto c\nabla F$, and the free energy function across the membrane is $F = RT\ln c + U$, where $U$ models an energy input by the cell. The solute is transported via two different types of channels: For passive ion channels, $U = 0$ and there is no energy input. For active ion pumps, $U \neq 0$, and the energy difference across the membrane, $G_a = -\Delta U = (U_2 - U_1)$ in Fig 1c, is the driving force of the active solute transport. This energy input usually comes from ATP hydrolysis. In our model, we assume that a fraction of the energy derived from ATP hydrolysis is utilized for active pumping. Thus, in our model, a single parameter, $G_a$, describes the active solute transport properties of the cell.

Inside the membrane domain, the solute flux is continuous. Adding the fluxes through both types of solute carriers together, we obtain the total flux: $J_s = \gamma\tilde{G}_a\Pi - \eta\Delta\Pi_m$. Here $\tilde{G}_a = G_a/RT$ and $\Delta\Pi_m$ is the osmolarity difference across the membrane. $\gamma$ is the active ion transport coefficient, and $\eta$ is defined as the sum of active and passive ion transport coefficients (see S1 Text). Similarly, the water flux across the cell membrane is proportional to the free energy difference of water across the membrane: $J_w = \alpha(\Delta P - \Delta\Pi - \Delta\Pi_p)$ [19–21]. Note that the macromolecules ($\Pi_p$) contribute to water flux $J_w$ but do not directly contribute to the solute flux $J_s$. Also, the solute flux depends on the total osmolarity $\Pi$, not just the osmolarity difference $\Delta\Pi_m$. This is a direct result of the free energy function across the membrane, and will have important implications for systemic flux and pressure later. In the following, when not specified, the solute osmotic pressure ($\Pi$) only refers to that of the permeable small molecules.

The parameter $G_a$ describes the active solute transport property of the cell, which encapsulates the strength of individual ion pumps and the number of ion pumps embedded in the cell membrane. Experiments have shown $G_a$ is not constant but can depend on pressure. For example, when pressure is applied to the basal side of a kidney epithelium, sodium/potassium exchanger (NKE) is observed to leave the basal-lateral side, slowing the overall solute flux [3]. NaCl flux across the epithelium is the major driver of water flux and NaCl flux also has been shown to decline with increasing pressure [5]. Therefore, a natural assumption in the model is that cells reduce the solute driving force, $G_a$, when pressure at the basal side is increased. Similar arguments can be applied to osmotic pressure, although direct experiments are still needed. A simple model that incorporates this pressure/osmolarity sensing by cells is: $J_s = \gamma\tilde{G}_a\Pi - \eta\Delta\Pi_m - m\Delta P - m'(\Delta\Pi + \Delta\Pi_p)$, where $(a,b)$ denote apical and basal surface, $\Delta P =$

$P_b - P_a, \Delta\Pi = \Pi_b - \Pi_a, \Delta\Pi_p = \Pi_{p,b} - \Pi_{p,a}$ are the differences across the cell. Here we assume that the cells are sensing the total osmotic pressure contributed by both small solutes and impermeable macromolecules. $m$ and $m'$ describe solute transporter polarization as a function of pressure and osmolarity gradients across the membrane.

Once the water and solute fluxes across the apical and basal membranes are determined, the fluid and solute transport equations can be solved simultaneously across the epithelium to obtain the overall concentration profile and the pressure field. An excellent analytic approximation can be made (see S1 Text) and the results are

$$J_w = \alpha_{ss}\left[-\Delta P + \zeta_{w1}\Delta\Pi + \zeta_{w2}\Delta\Pi_p + \zeta_{w3}\Pi_0\right] \qquad (2)$$

$$J_s = \alpha'_{ss}\left[-\Delta P + \zeta_{s1}\Delta\Pi + \zeta_{s2}\Delta\Pi_p + \zeta_{s3}\Pi_0\right] \qquad (3)$$

where $J_w$ ($J_s$) is the water (solute) flux across the epithelium, respectively. $\Delta\Pi = \Pi_a - \Pi_b$, $\Delta\Pi_p = \Pi_{p,b} - \Pi_{p,a}$, $\Delta P = P_a - P_b$ and $\Pi_0 = \frac{(r+1)\Pi_a + \Pi_b}{r+2}$ is a weighted mean osmotic pressure of the permeable molecules. $r = \frac{\gamma}{\eta}\tilde{G}_a$ is the dimensionless energy input by the cell, which is the relative capacity of active ion transport compared to total ion transport. Note $r \ll 1$, and therefore $\Pi_0 \sim (\Pi_a + \Pi_b)/2$. $(\alpha_{ss}, \alpha'_{ss})$ are the two effective permeability coefficients which are functions of molecular parameters. $\zeta_{wi}, \zeta_{si} (i = 1, 2, 3)$ are also functions of cell/molecular parameters. In particular, $\zeta_{si}$ also depends on the mean osmotic pressure of permeable molecules ($\Pi_0$) (see S1 Text). Eqs (2) and (3) represent the so called pump performance curve (PPC), which describes how flux changes with external pressure. We can also include the paracellular (through junctions) water and solute fluxes, which only modify the coefficients in Eqs (2) and (3). Detailed discussions, including a glossary of parameters and their estimation process, are given in the supplemental material (S1 Text, S1 Table). Here, we refer to the epithelial/endothelial layer as an active pump. As can be seen from the derivation above, the active process is only the solute transport. The fluid transport follows pressure and osmolarity gradient and therefore is more of a passive process. Here the pressure and osmolarity gradient are given as input parameters. We will see later that the pressure and osmolarity of different physiological compartments are dynamic properties of the connected network model, which includes these pumping elements.

The computed solute osmolarity profiles in the cytoplasm are shown in Fig 2a–2b, which demonstrate how the cytoplasmic concentration profiles can adapt to the external osmolarity for a given energy input and pressure gradient, captured by the dimensionless cell energy input $\tilde{G}_a$ and $\Delta P$. As the external osmolarity and pressure gradients change, the cytoplasmic osmolarity profiles also vary following the external change. There is a critical osmolarity (pressure) difference such that the flux can be reversed. The energy input and other cell properties such as membrane permeability can be adjusted by the cell by changing polarization/gene expression. Eq (2) is consistent with the classic Starling's equation of endothelial leakiness: $J_w = \alpha(-\Delta P + \sigma\Delta\Pi)$ [22]. In the limit $\tilde{G}_a = 0$ and no osmolarity difference for permeable solutes ($\Delta\Pi = 0, \Delta\Pi_p \neq 0$), the predicted Starling coefficient is $\sigma = \zeta_{w2} = \frac{D + \eta L - 2m'L}{D + \eta L + 2mL}$.

The results for solute and water fluxes predict that cells can adapt to changing osmolarity and pressure conditions (Fig 2c–2d, S1 Fig). These fluxes are consistent with the idea of active "pumping", i.e., the flux declines with increasing hydraulic pressure difference: $J_w = -\alpha_{ss}(\Delta P - \Delta P_w^*)$ and $J_s = -\alpha'_{ss}(\Delta P - \Delta P_s^*)$, where $\Delta P_{s,w}^*$ are the stall pressures when fluxes reach 0. When the external gradients are too high, the flux can be reversed. Interestingly, we predict that the total background osmolarity, $\Pi_0$, has an effect on water and solute fluxes. Depending on the pressure and osmolarity gradient, the mean external osmolarity may

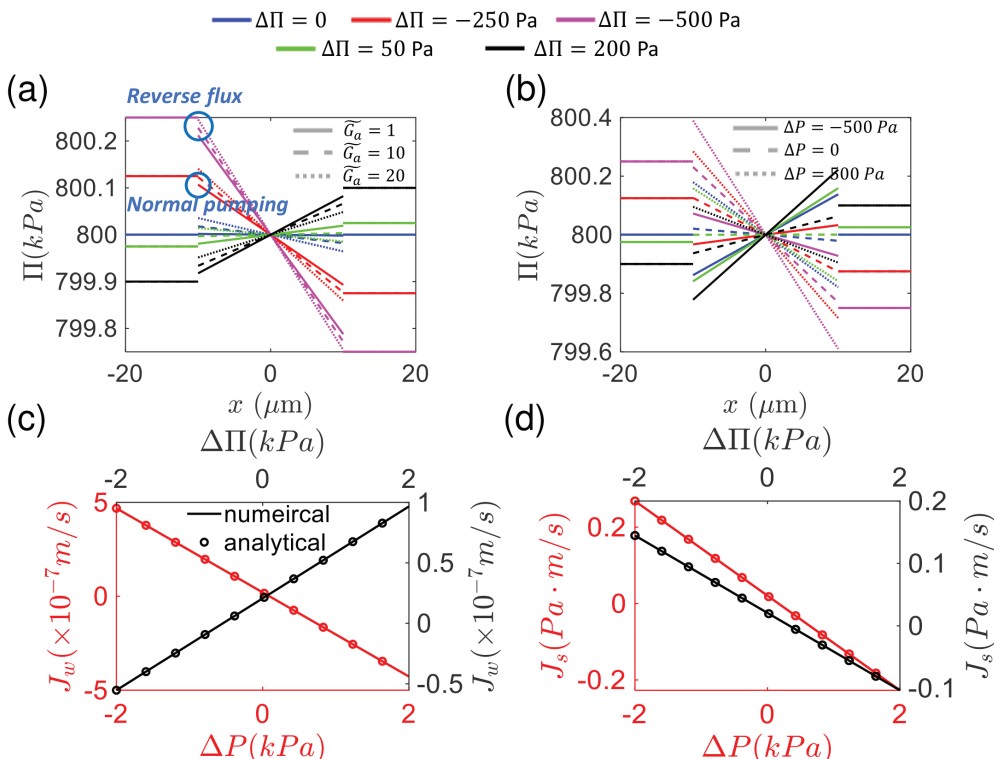

**Fig 2. Cytoplasmic osmolarity profile and transport fluxes under varying external osmotic and pressure conditions.** (a)–(b) Cytoplasmic solute concentration (osmolarity) profile when external osmolarity and pressures conditions are varied. The osmolarity profile in the cytoplasm adapts to external changes. Water flux decreases when external basal-apical osmotic pressure difference increases, and the transport direction could reverse. Increasing the energy input and the pressure gradient decrease the slope of the inner concentration profile. Here, the mean osmotic pressure of basal and apical sides are 800 $k$Pa, and only $\Delta\Pi$ is changed. In (a) and (b), $m = m' = 0$. (c)–(d) Comparisons between the accurate numerical solution and the analytic approximation for water and solute flux under varying external osmotic and pressure conditions. In the calculation, when not specified, $\Delta P$ and $\Delta\Pi$ are set as zero and the mean osmotic pressure is set as $\Pi_0 = 800$ $k$Pa.

have an opposite influence on the fluxes (Fig 3a–3b). The critical condition is derived in the supplementary material S1 Text. With a sufficiently high osmolarity, water flux is limited by the rate of active ion pumping ($\gamma$) and energy input ($\tilde{G}_a$). The limiting water flux when mean osmolarity reaches infinity is predicted to be $J_w = \gamma\tilde{G}_a$ by Eq S24. The influence of ion channel permeability ($\gamma, \eta$) and membrane water permeability ($\alpha$) is shown in S2 Fig.

Our model can also predict stall pressure/osmolarity (zero flux) as well as phase diagrams of when flux reversals can occur (Fig 3c–3f). The stall pressures are:

$$\Delta P_w^* = \zeta_{w1}\Delta\Pi + \zeta_{w2}\Delta\Pi_p + \zeta_{w3}\Pi_0 \qquad (4)$$

$$\Delta P_s^* = \zeta_{s1}\Delta\Pi + \zeta_{s2}\Delta\Pi_p + \zeta_{s3}\Pi_0 \qquad (5)$$

The stall pressures depend on external osmolarity and ion pump relocalization. Pressure or osmolarity changes will result in cell polarization change and ion pump re-localization. This is described by coefficients $m$ and $m'$ (included in coefficients $\zeta_{wi}, \zeta_{si}$). Note that epithelial mechanical integrity may also ultimately determine stall pressure, but this is not captured here. In the results above, $\Delta\Pi$ refers to permeable solute osmolarity difference

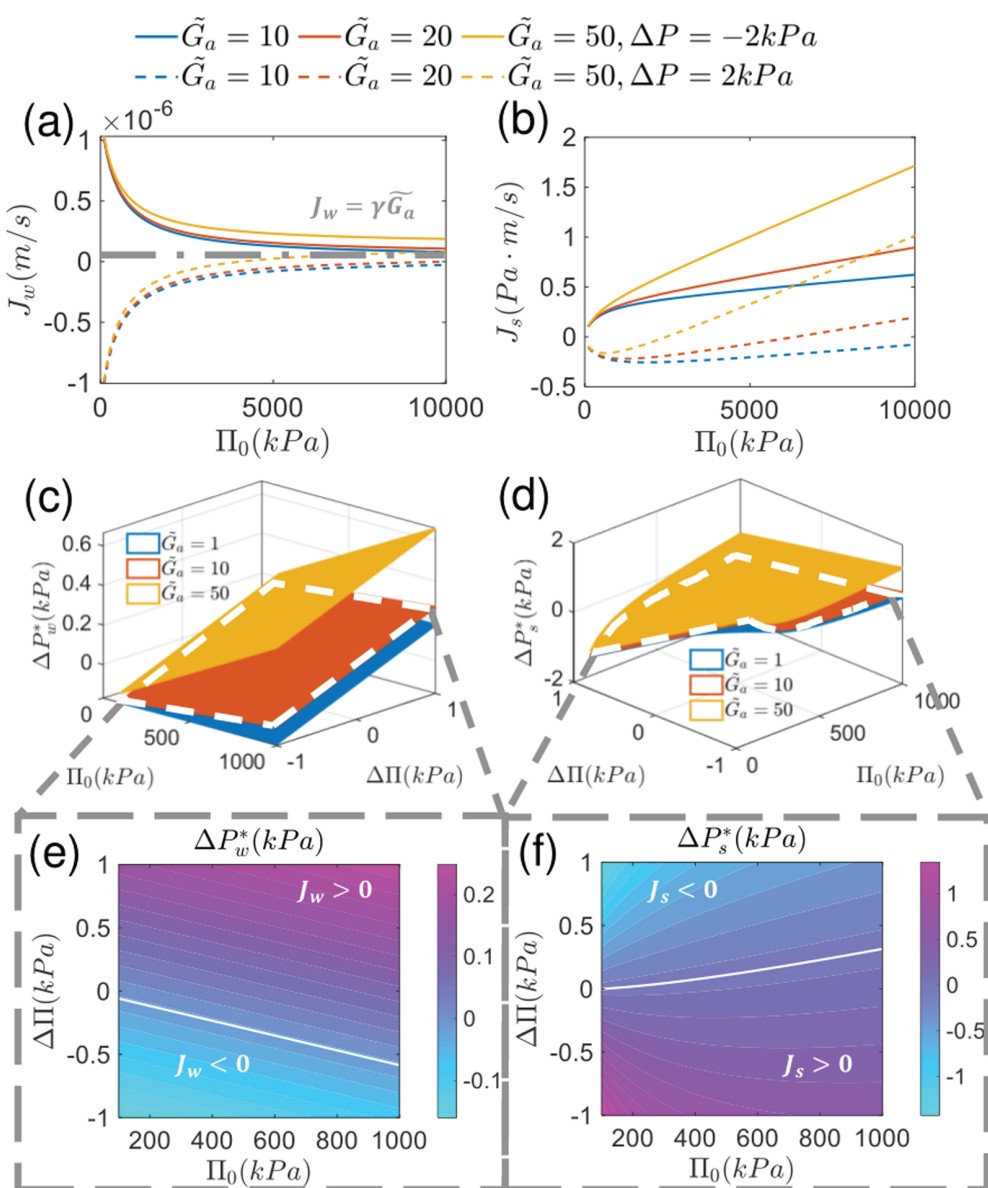

**Fig 3. Water and solute transport as a function of absolute value and gradient of osmotic pressure and energy input.** (a)–(b) Influence of mean background osmotic pressure $\Pi_0$ on water and solute flux. There are two branches in the plot corresponding to different pressure gradients. There is a limit for water flux ($J_w \approx \gamma \tilde{G}_a$) when the osmolarity $\Pi_0$ is sufficiently high. (c)–(d) Dependence of stall pressure for both water ($\Delta P_w^*$) and solute ($\Delta P_s^*$) on absolute osmolarity ($\Pi_0$), osmolarity gradient ($\Delta\Pi$) and energy input ($\tilde{G}_a$). (e)–(f) Phase diagram of water and solute transport. The boundary is the stall pressure, across which the transport direction reverses. For the water flux, the pump functions normally above the boundary line and "reverses" below the boundary line. For solute flux, the region below is the normal regime. As the basal-apical pressure difference ($\Delta P$) increases, reversed flux becomes more likely for both water and solute.

across the pump. The osmolarity difference of impermeable macromolecules is set as zero ($\Delta\Pi_p = 0$). The effect of $\Delta\Pi_p$ on water and solute flux across the pumping element is shown in S3 Fig.

## Connected microphysiological organ systems: A model of fluid circulation

In the previous section we discussed an isolated pumping unit. In an organism, epithelia and endothelia are components in a complex circulatory system. The overall fluid circulation flux is a systemic property with central importance in physiology and medicine. Therefore it is of great interest to examine the effect of "pumping" on the circulatory network from a systemic viewpoint. Modern bioengineering are developing microphysiological systems, where microfluidics are combined with organoid culture to study biological transport and function [23–25]. These systems allow for microscale control of pressure and osmolarity, which can reveal cell-driven transport properties. There are also many simulation studies on the human circulatory system using lumped parameter models [26–29]. In analogy to electrical circuits, models such as the Windkessel model treat fluxes in blood vessels using concepts of resistors and capacitors [30]. Despite detailed modelling of pressures in the network, the influence of osmotic pressure and solutes is generally neglected. Moreover, no "pumping" elements are included, which significantly alters the local and global properties of the circulatory system. Here, we will use the lumped parameter approach, but develop a physiological model that includes possible epithelial/endothelial pumping units. We seek to understand local and global influence of the pump on the overall network. We also explore the influence of osmolarity and energy input of the pump on the overall circulation.

The model consists of three elements: the blood vessel (mainly aorta and vena cava), capillary blood vessels in the organs (both resistance and compliance are considered) and the pumping element (epithelial and endothelial tissues). The pumping element is modeled using water and solute fluxes given in Eqs (2)–(3). All blood vessels (aorta, vena cava and organ capillaries) are described by a two-element Windkessel model, which is a resistor-capacitor set in parallel.

### A one-pump circulation model

We first examine a one-pump circuit model in which the pumping element incorporates the epithelial layer, endothelial layer, and interstitium. The inlet of the pump is lumen side of the renal tubule and the outlet faces the lumen of blood vessels. An illustration of the model is shown in Fig 4a. Blood is pumped from the heart (modeled as a source with pressure $P_s$) and goes through aorta (with resistance $R_A$), and branches before the kidney. A portion of this flow goes through the capillary blood vessel of the organs (effectively modeled as a resistor $R_O$) while others are filtered by the glomerulus ($R_G$) and then goes through the pumping unit consisting of epithelium, interstitium and endothelium (water reabsorption). The two branches finally merge and return to the heart through the vena cava ($R_V$). For convenience, we define the node pressure at the end of vena cava to be zero. All the capacitors can be combined together as one effective capacitor $C_E$, which gives trivial prediction on dynamical response of the pressure and flux. Therefore, we will neglect "capacitors" and focus on the static property of the system.

Similarly, we can define the circuit of osmolytes, in which the hydraulic pressure is replaced by osmotic pressure (Fig 4b). The osmotic pressure remains the same through the blood capillaries in organs but changes across the glomerulus. The plasma proteins are not filtered through the glomerulus and therefore create an oncotic pressure difference $\Pi_P$, which is approximately $\Pi_{p,1} - \Pi_{p,2} = \Pi_P = 3.8 kPa$ [31]. In the one-pump model, the osmotic pressures of permeable solutes are equal at all nodes.

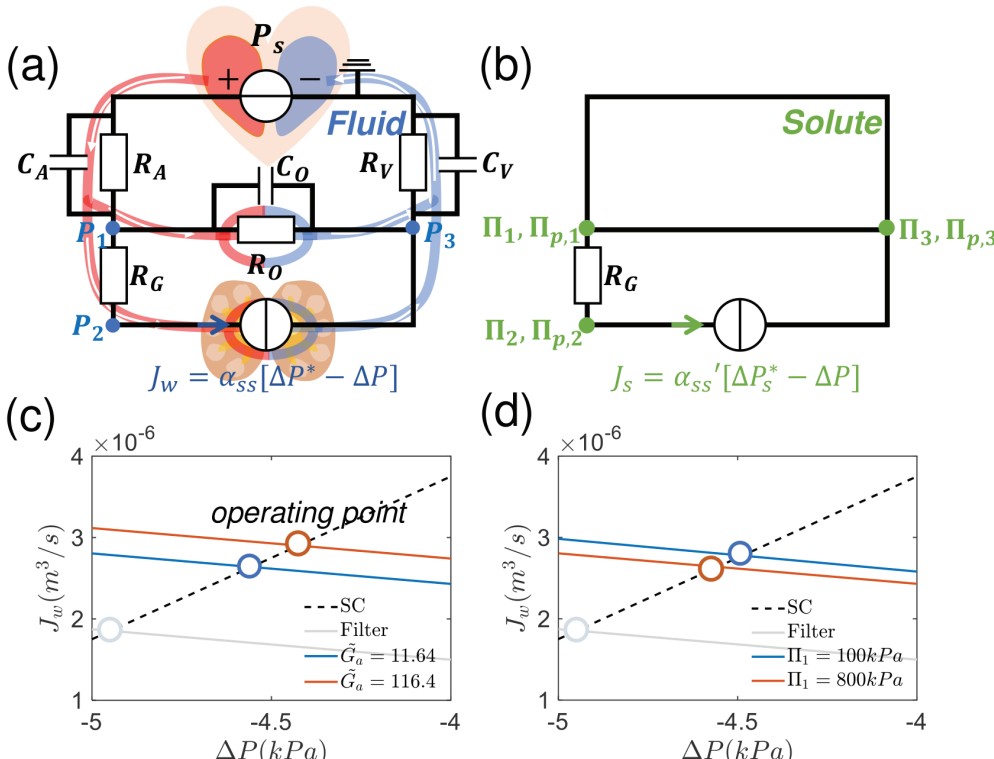

**Fig 4. Schematic of the one-pump circulatory system and its solute circuit with pump performance curves.** (a) Diagram of the one-pump circulatory system. The circulatory system is simplified into 6 parts: heart aorta, vena cava, organs, glomerulus and the pumping element including renal tubule epithelial tissue and endothelial tissues. Each part is modeled as a two-element Windkessel element. (b) The solute circuit is considered simultaneously with the fluid circuit. The total osmotic pressure ($\Pi + \Pi_p$) is different across the glomerulus $R_G$ ($\Pi_1 = \Pi_3 \neq \Pi_2$). (c)–(d) The systems curve, pump performance curve and the operating point. Increasing the cell energy input elevates flux and pressure across the pump while the external osmotic pressure decrease flux and pressure. When the pumping element is replaced by a passive filter ($\tilde{G}_a = 0$, both flux and pressure gradient decrease significantly. In (c), the oncotic pressure ($\Pi_p$) in the blood plasma is fixed and the osmotic pressures at different nodes always satisfy: $\Pi_1 = \Pi_2 = \Pi_3$, $\Pi_{p,1} = \Pi_{p,3}$, $\Pi_{p,1} - \Pi_{p,2} = \Pi_p$.

Just as in an electrical circuit, we can use Kirchhoff's law to compute pressures at different nodes. However, different from Ohm's law, the water flux through an element (e.g., capillaries in the organ, pumping element) is a function of both hydraulic pressure and osmotic pressure differences across the element, which is: $J_w = (\Delta P - \Delta\Pi - \Delta\Pi_p)/R$, where $R$ is the "effective resistance" of the element. Osmolarity is constant in the vessels and only changes across the glomerulus and the pumping element. Consequently, the solute flux in vessels is only proportional to the water flux. The solute flux through a pumping element is given by Eq (3). There are three nodes in the network and the equations are:

$$\frac{P_s - P_1}{R_A} = \frac{P_1 - P_3}{R_O} + \frac{P_1 - P_2 - \Pi_P}{R_G} \tag{6}$$

$$\frac{P_1 - P_2 - \Pi_P}{R_G} = \alpha_{ss}\left[-(P_3 - P_2) + \Delta P^*\right]S_{Ep} \tag{7}$$

$$\alpha_{ss}\left[-(P_3 - P_2) + \Delta P^*\right]S_{Ep} + \frac{P_1 - P_3}{R_O} = \frac{P_3}{R_V} \tag{8}$$

where $R_A, R_O, R_G, R_V$ are resistances of aorta, capillaries in organs, glomerulus and vena cava, respectively. $P_s$ is the pressure heart generates and $P_1, P_2, P_3$ are the node pressures. $\Pi_1, \Pi_3$ are osmotic pressure at nodes 1 and 3. The pumping performance of the kidney cells is: $I = \alpha_{ss}[\Delta P^* - (P_3 - P_2)]$, where $\alpha$ is a permeability constant and $\Delta P^*$ is the "stall pressure" given by Eq (4). The coefficient $S_{Ep}$ is the total surface area of the renal tubule, which transforms the velocity (m/s) into the volume flow rate (m³/s). By defining $R_O = R, R_A = k_1 R, R_G = k_2 R, R_V = k_3 R$, we can get general analytical solutions for all the node pressures, total flux and branch flux (S1 Text).

An important aspect of the circuit is the systems curve, which determines the flux and pressure drop across the pump. The intersection of the pump performance curve and the systems curve is the operating parameters of the pump when it is placed into the circuit. The systems curve is derived from Eqs (6)–(8):

$$I_{23} = \frac{1}{R} \frac{(k_1 + k_3 + 1)(\Delta P - \Pi_P) + P_s}{k_1 + k_2 + k_3 + k_1 k_2 + k_2 k_3} \tag{9}$$

Fig 4c–4d show the systems curve and the PPC with different energy input ($\tilde{G}_a$) and the osmolarity of the permeable solute ($\Pi_1$). With increased energy input, the pump performance curve shifts upward, resulting in increased water flux. With increase $\Pi_1$, however, the systemic flux decreases. When active pumping is removed ($\tilde{G}_a = 0$), both the flux and pressure difference decrease significantly. The results are intuitive, but we see that systemic properties such as resistances in various vessels will impact flow and pressure across the epithelium.

We systematically examine how energy input ($\tilde{G}_a$), blood oncotic pressure ($\Pi_p$), external osmotic pressure of permeable solutes ($\Pi_0$), and membrane permeability to water and solutes ($\gamma, \eta, \alpha$) influence node pressures and water flux (S4 Fig, S5 Fig). In particular, the oncotic pressure results provide some explanations of kidney diseases related to protein filtration defects. If the glomerulus cannot block blood proteins, osmotic pressure is increased at node 2 and changes water flux and pressure distribution.

## A two-pump circuit model including the interstitium

In the previous section, epithelial tissue and endothelial tissue are combined together with the interstitium to form an equivalent pump. In reality, there are osmolarity and pressure differences across epithelial/interstitial/endothelial compartments, generated by active solute transport [32,33]. Osmotic and hydraulic pressure gradients in these three compartments potentially play an important role in the overall water transport. Currently, there is no concrete evidence showing active fluid pumping in endothelial cells. Therefore, our assumption that endothelial cells are active pumps is speculative. However, studies in corneal endothelial cells show that this endothelium can actively pump ions to adjust its environmental osmolarity [34,35]. Therefore, the transport of water following pressure and actively established osmolarity gradient is possible, which also rationalizes our assumption that the endothelial cells can actively pump fluid. In this section, we explore a more detailed model which includes the epithelial pump, possible endothelial pump, and the interstitium. Fig 5a–5b show the circuit in consideration. Node 3 corresponds to the interstitium. We solve the coupled system that includes both the hydraulic pressure circuit and osmotic pressure circuit. Using Kirchoff's law, pressures at four nodes are:

$$\frac{P_s - P_1}{R_A} = \frac{P_1 - P_4}{R_O} + \frac{P_1 - P_2 - \Pi_P}{R_G} \tag{10}$$

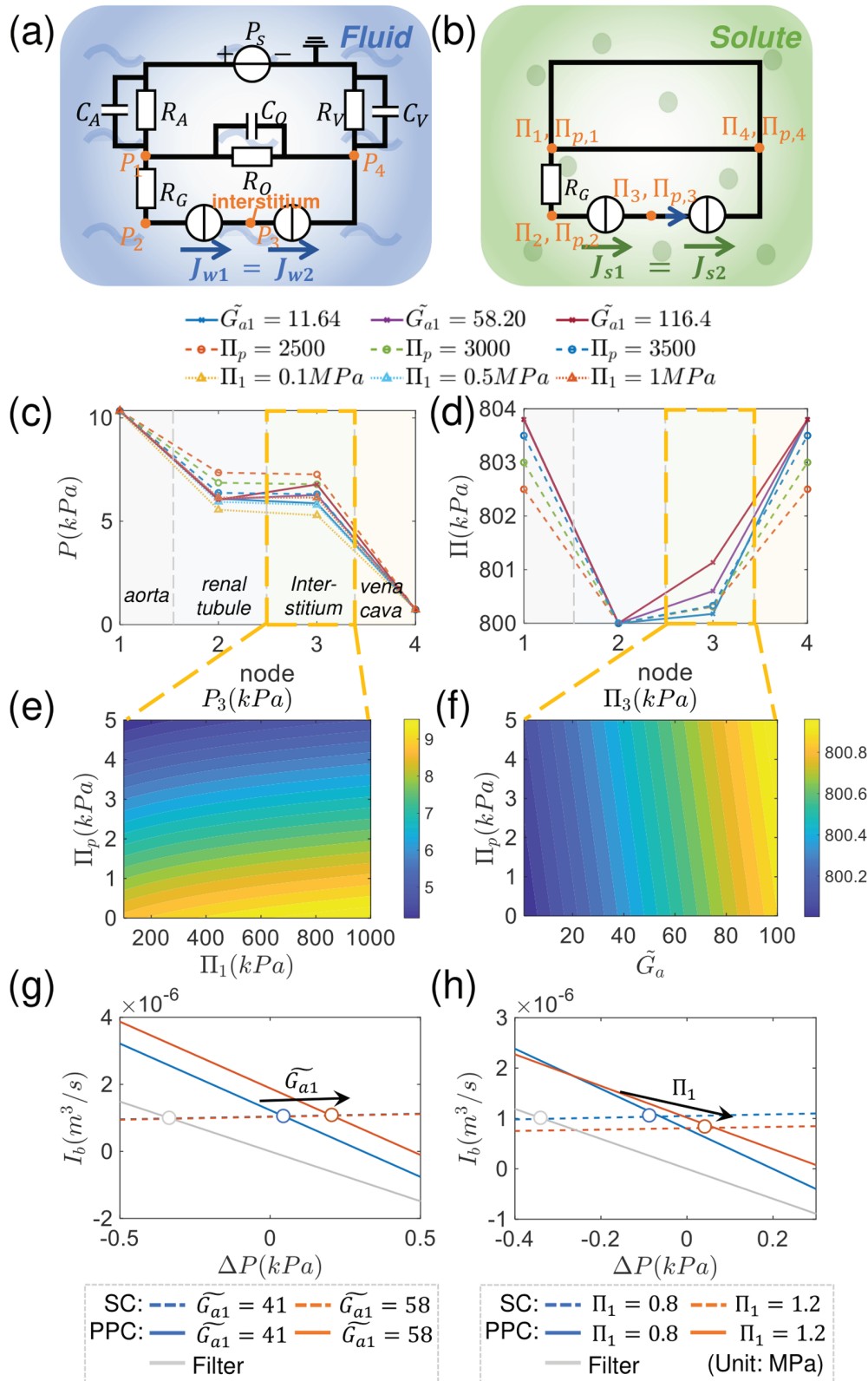

**Fig 5. Diagram of the two-pump water and solute circuit with spatial distribution of pressures and osmotic gradients.** (a)–(b) Diagram of the water and solute circuit for the two-pump model. (a) Fluid circuit. Different from the one-pump model, the epithelial tissue, endothelial tissue and interstitium are all included. (b) Solute (osmolarity) circuit. The osmotic pressures are the same across the organs (node 1 and 4). There is an osmolarity gradient after the

**Fig 5.** glomerulus since proteins are filtered. The osmotic pressure difference includes oncotic pressure. Osmotic pressure at node 3 may also be different, which is determined by the solute flux balance. (c)–(d) Spatial distribution of pressure and osmolarity. (e)–(f) Influence of total blood osmotic pressure, oncotic pressure and energy input on hydraulic pressure and osmotic pressure in the interstitium.(g)–(h) The systems curve, the pump performance curve, and the operating point corresponding to the "epithelial pump". In all calculations, when not specified, the energy inputs for kidney epithelial pump and endothelial pump are: $(\tilde{G_{a1}}, \tilde{G_{a2}}) = (29.10, 11.64)$. All other parameters are the same for both pumps.

$$\frac{P_1 - P_2 - \Pi_P}{R_G} = \alpha_{ss1}\left[-(P_3 - P_2) + \Delta P_1^*\right]S_{Ep} \tag{11}$$

$$\alpha_{ss1}\left[-(P_3 - P_2) + \Delta P_1^*\right]S_{Ep} = \alpha_{ss2}\left[-(P_4 - P_3) + \Delta P_2^*\right]S_{Ed} \tag{12}$$

$$\alpha_{ss2}\left[-(P_4 - P_3) + \Delta P_2^*\right]S_{Ed} + \frac{P_1 - P_4}{R_O} = P_4/R_V \tag{13}$$

where $\Delta P_1^*, \Delta P_2^*$ are stall pressures for epithelial and endothelial cells, respectively. To obtain a solution, another equation describing solute flux balance in the interstitium is needed ($J_{s1} = J_{s2}$), which can be obtained from Eq (3).

We numerically solve the equations and analyze the spatial distribution of pressure and osmolarity in the circuit. We are interested in how the external mean osmolarity and ion pump energy input influence the total water flux and the flux across the pump. Node 1-4 correspond to the glomerulus, the renal tubule, the interstitium and the vein, respectively. In general, the pressure drops from the glomerulus to the vein. However, the pressure in the interstitium (node 3) is higher than that in renal tubule (node 2) and vein (node 4). The pressure in interstitium can be further elevated by increasing the epithelial pump energy input ($\tilde{G}_{a1}$), decreasing the blood oncotic pressure ($\Pi_p$), or increasing the external osmolarity of permeable solutes (e.g., $\Pi_1$) (Fig 5c). When varying the osmotic pressure of permeable molecules, the interstitial hydraulic pressure can vary from 5 kPa to 9 kPa (Fig 5e). In contrast to the hydraulic pressure, the total circuit osmolarity (small molecules and macromolecules) increases monotonically from node 2 to node 4. The osmolarity in the interstitium is in between that of renal tubule and renal vein. The energy input $\tilde{G}_{a1}$ increases the osmotic pressure of the interstitial fluid due to increased ion transport into the interstitium. However, blood oncotic pressure ($\Pi_p$) has little effect on the osmotic pressure in the interstitium (Fig 5d). The interstitial osmolarity can vary up to 0.5 kPa with varying blood oncotic pressure and energy input (Fig 5f).

Another interesting quantity is the total circulatory water flux. When the mean osmolarity of permeable solutes outside the pump ($\Pi_0$) is increased, both the total water flux and the branch flux across the pump decrease. As expected, increasing the energy input, $\tilde{G}_a$, increases the total flux and branch flux. Increasing the blood oncotic pressure (consequently the osmotic pressure difference between node 2 and 4) decreases both the total flux and branch flux across the pump (S7 Fig). The branch flux across the pumps can also be determined by locating the intersection between the system curve and the pump performance curve. In the two-pump model, it is not feasible to derive separate systems curves for each pump. Consequently, this section focuses solely on exploring the system curve and pump performance curve corresponding to the epithelial pump. Similar to the one-pump model, an increase in energy input shifts the operating point upward, resulting in larger flux and pressure difference (Fig 5g). However, unlike the one-pump model, external osmolarity changes both the systems curve and pump performance curve (Fig 5h). When active pumping is

stopped, both the pressure difference and flux decrease (Fig 5). In the supplemental material (S9 Fig), we also explore the case where the endothelial layer is a passive filter. Results show minimal differences as compared to the scenario where the endothelium is modeled as an active pump. Additional results on the influence of membrane permeability to water and solutes are provided in S6 Fig, S8 Fig. It is important to note that, unlike the one-pump circuit, the systems curve here also incorporates information from the "endothelial pump" and the osmotic pressure of permeable solute at node 3, since all elements are coupled together to generate the total circulatory flux.

An interesting prediction in this model is that both the spatial variation and the absolute value of osmotic pressure influences the overall circulation and vessel flux. When the mean external osmotic pressure is increased, water flux decreases. In order to maintain the overall flux at the same level, the pressure from the heart needs to be elevated or blood vessels need to constrict. This can be interpreted as one possible mechanism of hypertension. Another approach for maintaining the water flux is by increasing the energy input and ion pump (e.g. NaK) activity, which can be controlled by hormones and other physiological inputs. Indeed, all parameters in the model are controlled by physiological response, e.g., vessel constriction from smooth muscle action. Therefore, in order to model physiological fluid circulation, additional layer of feedback control is needed. It is likely that hydraulic pressure and osmolarity are constantly sensed by cells in various compartments, and feedback control is used to maintain systemic homeostasis.

## Mechanical efficiency and stress in the pumping element

It is worth noting that for most mechanical pumps, there exists a regime of optimal efficiency and minimal mechanical stress. Similar regimes are likely to exist for cells. If the operating point deviates from this optimal regime, cells are likely to experience increased metabolic and/or physiological stress, potentially predisposing them to various diseases. For example, when hydraulic pressure gradient is altered across the epithelium, cell proteome change was observed [3]. Increased pressure gradient can challenge epithelial integrity and disrupt junctions [36]. We can explore these questions by examining energy efficiency of the pump. In the following discussion, we assume $\Delta\Pi = \Delta\Pi_p = 0$ across the pump. The relative mechanical efficiency can be defined as: $\epsilon = \xi J_w^2/(G_a - m\Delta P/\gamma\Pi_0)$, where $\xi$ is an effective parameter that includes the surface area of epithelial layer, ATP concentration, ATP hydrolysis rate, and the fraction of ATP hydrolysis energy allocated to active ion pumping, etc. It is important to note that, in this formulation, we assume the energy consumption for active ion pumping is a fixed fraction ($\lambda$). However, in biological contexts, $\lambda$ is likely to depend on environmental factors such as the ATP hydrolysis rate, hydrostatic pressure gradient, and osmotic pressure gradient. A more comprehensive ion transporter model that incorporates these dependencies would be a valuable direction for future work. With increasing basal-apical pressure difference ($\Delta P$), the energy efficiency decreases and eventually reaches 0 at stall pressure $\Delta P^*$, where cells experience the maximum mechanical stress. The energy curve also depends on energy input for solute transport ($\Delta\tilde{G}_a$) and ion transporter relocalization ($m$) (Fig 6a). In order to achieve optimal energy efficiency and alleviate mechanical stress, cells may potentially adjust the pumping parameters (e.g., water permeability $\alpha$) to modulate the pump performance curve. Additionally, the system curve could be altered in response to signals from stressed cells (Fig 6b). These possible adaptive mechanisms collectively maintain optimal energy efficiency and minimal mechanical stress. If this control system is disrupted, such as mutations observed in PKD [37], diseases may result.

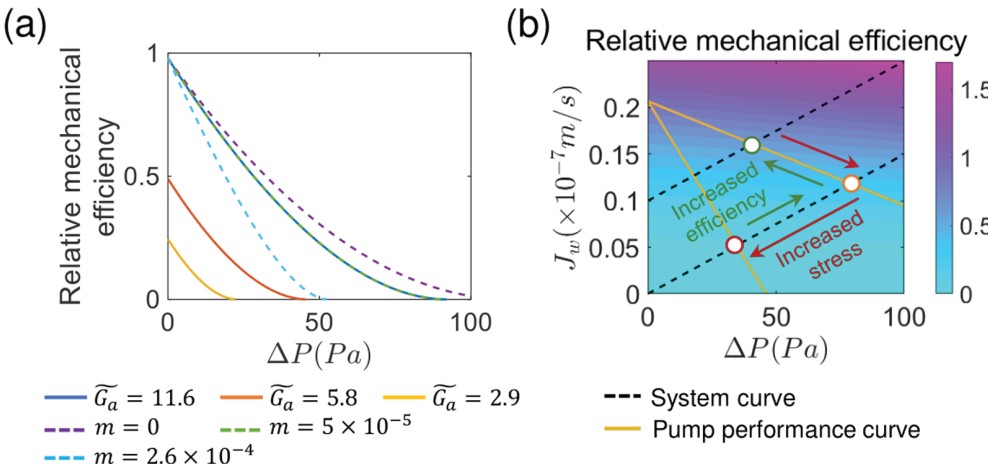

**Fig 6. Mechanical efficiency and pump performance curves.** (a) Relative mechanical efficiency as a function of energy input ($\Delta \tilde{G}_a$) and pressure dependence of solute flux ($m$). (b) Pump performance curves and energy efficiency. Cells could potentially modify both the PPC and the systems curve to enhance mechanical efficiency.

## Discussion and conclusion

In this paper, we develop a simple physical model explaining the "pumping" property of renal epithelial and endothelial cells. The model incorporates both solute pumping and diffusion, demonstrating that water transport is facilitated by the establishment of a solute concentration gradient in the cytoplasm. We start with the solute convection-diffusion equations. The equations are solved together with equations of water flux. Our model shows that the cytoplasmic solute concentration profile follows the external osmolarity change. To ensure normal water flow direction (from apical to basal), the cell actively transports solutes, maintaining a higher solute concentration on the apical side and a lower concentration on the basal side, relative to the external concentration.

Our model allows us to explore the influences of cell energy input on the generalized "pump performance surface" (water flux vs. pressure gradient and osmolarity gradient) of both water and solute. As expected, water flux decreases with an increase in the basal-apical pressure difference, and increases with the increase of basal-apical osmotic pressure difference $\Delta \Pi$. The solute flux $J_s$ decreases with both $\Delta \Pi$ and $\Delta P$. Interestingly, the absolute value of the external osmolarity also influences the water and solute flux. There are two different regimes where the absolute osmolarity has opposite influences on the flux. The boundary is a function of both pressure and osmolarity gradient. The energy input for ion pump also increases the water and solute flux. We also predict how the osmolarity gradient and the total osmolarity influence the stall pressure for both water and solute. We are able to obtain a phase diagram of water and solute flux, showing regimes of normal pumping and "reversed pumping". We find that when the energy input goes to zero, we obtain the Starling relation for the permeabilities of epithelia and endothelia.

The cell pumping model serves as the starting point for a realistic physiological circulation model. Previous models of systemic circulation have primarily focused on fluid fluxes, while neglecting the effects of solutes. The novel feature of our work is that the overall fluid circulation is coupled to solute circulation. This integration allows us to calculate systemic properties such as the systems curve (flux vs. pressure gradient in a systematic view) and pressure/osmolarity at various points in the network. By combining the system curve with

the pump performance curve, we can determine the operating point of the pumping element. Notably, for most mechanical pumps, there exists a regime of highest efficiency/lowest mechanical stress. A similar regime is likely to exist for cells. If the operating point is outside of this regime, cells may experience stress, which could lead to disease. We also explored how blood oncotic pressure and pump energy input influence the pressure distribution and water flux. The one-pump model predicts that both the total flux and branch flux across the pump decrease with increasing external osmolarity. Additionally, the results regarding how blood oncotic pressure influences flux offer insights into kidney diseases related to protein filtration [38,39].

A more realistic physiological model consists of two pumps and an interstitium. Again, water and solute circuits are solved together. This model shows a non-monotonic change in pressure across the network – the interstitial pressure is higher than that in the renal tubule and renal vein. The interstitial pressure is influenced by pump energy input, blood oncotic pressure and total blood osmolarity. The predicted osmolarity monotonically increases from the renal tubule to the vein and the osmolarity in interstitial fluid is increased with the increase of ion pump energy input.

Similar to the one-pump model, the two-pump circulation model predicts that both the total flux and branch flux across the pump decrease with the external osmolarity. This result provides a potential explanation for hypertension. Since the circulation flux decreases with increasing osmolarity, the body has to increase the blood pressure to maintain the same circulation flux. Note that our model does not contain possible physiological control of all circulation parameters. In reality, there are multiple ways that pressure, osmolarity, and flux are sensed. Hormone and neural signals can change all parameters in the circuit via feedback mechanisms. For example, a well known feedback mechanism in kidney transport is the Tubuloglomerular feedback (TGF), which is a negative feedback regulation of glomerular filtration rate based on the concentration of sodium chloride in the distal tubule [40,41]. The active feedback control of overall circulation is not modeled here. However, our work incorporating the coupling between hydrostatic pressure and osmotic pressure circuit provides a starting point where mechanisms of active control can be assessed.

Results from the two-pump circuit model is different from those of the one-pump model. This demonstrates that properties of the circulatory network have a significant impact on the behavior of cells. For tissues embedded in the system, the pump performance should be solved together with other system parameters. The model can be made more realistic by incorporating details of additional ionic species, such as $K^+$, $HCO_3^-$, $H^+$, etc. Furthermore, including the effects of electric fields in cells and tissues [42–44], which are known to be important for tissue mechanics and morphogenesis, will also expand the range of predicted phenomena. We note that this coupling between cells and the overall circulation system is present in mature organisms as well as during embryo and organ development. Since the overall circulation network influences cell behavior, it is likely that overall circulation can influence cell phenotype specification.

Finally, our current model assumes fixed cells and tissue geometry. Pressure and osmolarity gradients exist across various compartments in the circulation network. This means that there is mechanical stress across the compartment boundaries, which could also impact cell behavior. Pressure gradients will generate deformation and tension in compartment walls. It is known that cell proliferation and death are influenced by mechanical forces [45,46]. Moreover, if the compartment walls and boundaries are allowed to deform, and cells are allowed to proliferate and differentiate, then we will arrive at a model of morphogenesis where the growth of organs and tissues are coupled to cell mechanical tension/stress, pressure gradients and fluid flux [47]. Indeed, a concrete (beyond phenomenological) model of morphogenesis

should include active fluid, nutrient, and $O_2$ circulation, as well as the impact of mechanical stress on cell behavior. Our model provides a starting point to examine morphogenesis and development in realistic physiological contexts.

## Supporting information

**S1 Text. Model details.** Detailed derivation of tissue-level fluid transport and systems-level fluid circulation.
(PDF)

**S1 Table. Model parameters.** Parameter values and sources for tissue-level and systems-level fluid transport models.
(PDF)

**S1 Fig. Generalized Pump Performance and Flux Regulation.** The water and solute fluxes of an isolated pump are determined by basal-apical pressure ($\Delta P$) and osmolarity differences ($\Delta \Pi$), energy input ($\tilde{G}_a$), and the sensitivity of solute flux to pressure and osmolarity gradients $m, m'$. The energy input $\tilde{G}_a$ increases both the water and solute flux (a&d). In the generalized pump performance surface, $m, m'$ decrease the slope of the fluxes with respect to pressure gradient $\Delta P$ and osmolarity gradient $\Delta \Pi$ (b,c,e,f).
(PDF)

**S2 Fig. Influence of active, passive ion transport coefficients ($\gamma, \eta$) and water permeability of the membrane ($\alpha$) on water (a–c) and solute flux (d–f) for an isolated pump.** (a) Active transport of ion increases the water flux. (b) Increasing passive transport coefficient decreases water flux. (c) Increase in water permeability leads to increased water flux, reaching a plateau. (d–f) Results on solute flux.
(PDF)

**S3 Fig. Effect of macromolecule osmotic pressure gradient on fluid transport.** Water and solute fluxes increase with the osmotic pressure gradient $\Delta \Pi_p$, highlighting the role of macromolecule-induced osmotic forces in driving transport across the membrane.
(PDF)

**S4 Fig. Influence of osmotic pressure on node pressure and blood flux with different energy input for ion pump ($\tilde{G}_a$, a-e) and blood plasma oncotic pressure ($\Pi_p$, f-j).** (a)–(c) Increase of the external osmotic pressure causes decrease in pressure at node 3 and increase in node 1 and 2. Increase of energy input decreases the pressure at node 1 and 2 while increases the pressure at node 3. (d)–(e) Total blood flux and the branch flux across the pump both decrease with external osmolarity while increase with energy input. (f)–(h) Oncotic pressure in blood plasma increases the pressure at node 1 while decreases the pressure at node 2 and 3. (i)–(j) Total blood flux and the branch flux across the pump both decrease with the increase of blood oncotic pressure. In (f)–(j), the energy input is set as: $\tilde{G}_a = 11.64$. The water transport constant is set as $\alpha = 5 \times 10^{-11} \ m \cdot s^{-1} \cdot Pa^{-1}$.
(PDF)

**S5 Fig. Influence of ion transport and water permeability on pressure and flux in a one-pump network.** The effects of active, passive ion transport coefficients ($\gamma, \eta$) and water permeability of the membrane ($\alpha$) on pressure distribution (a–c), total blood flux (d–f) and branch flux (g–i) across the pumping element for the one-pump network are explored. (a)–(c) Both the active ion transport coefficient ($\gamma$) and water permeability ($\alpha$) of the pump decrease the pressure in the lumen of the renal tubule (node 2). (d)–(i) Both the total flux and branch flux across the pump increase with $\gamma, \eta$, and $\alpha$.
(PDF)

**S6 Fig. Influence of ion transport and water permeability on pressure and osmolarity distribution in a two-pump network.** (a)–(c) Influence of active, passive ion transport coefficients ($\gamma, \eta$) and water permeability of the membrane ($\alpha$) on pressure distribution. The interstitial pressure (node 3) increases with the increase of active ion transport coefficient $\gamma$ and decrease of passive transport coefficient $\eta$ and water permeability $\alpha$. (d)–(f) Results on osmolarity distribution for the two-pump network. The interstitial osmolarity (node 3) increases with the increase of active ion transport coefficient $\gamma$ and decrease of passive transport coefficient $\eta$.
(PDF)

**S7 Fig. Influence of osmotic pressure on total flux and branch flux across the pumps with different ion pump energy input and blood oncotic pressure.** According to the two-pump model, both the total flux and branch flux across the pumping element decrease with elevated external osmotic pressure. (a)–(b) Increase of energy input for the ion pump increases the blood flux. (c)–(d) Increase of blood plasma oncotic pressure decreases the blood flux.
(PDF)

**S8 Fig. Influence of ion transport coefficients and water permeability on blood flux in a two-pump network.** The effects of active, passive ion transport coefficients ($\gamma, \eta$) and water permeability of the membrane ($\alpha$) on total blood flux (a–c) and branch flux (d–f) across the pumping elements for the two-pump network.
(PDF)

**S9 Fig. Effects of total blood osmotic pressure, oncotic pressure and energy input on hydraulic pressure and osmotic pressure when endothelial cells are not actively pumping** ($\tilde{G}_{a2} = 0$)**.** (a)–(b) spatial distribution of pressure and osmolarity with different energy input, blood oncotic pressure and total osmolarity in blood plasma. (c)–(d) Influence of total blood osmotic pressure, oncotic pressure and energy input on hydraulic pressure and osmotic pressure in the interstitium. When not specified, the energy inputs for kidney epithelial pump and endothelial pump are: $(\tilde{G}_{a1}, \tilde{G}_{a2}) = (29.10, 0)$. All other parameters are the same for both pumps.
(PDF)

## Author contributions

**Conceptualization:** Yufei Wu, Morgan A Benson, Sean X. Sun.

**Data curation:** Yufei Wu, Morgan A Benson, Sean X. Sun.

**Funding acquisition:** Sean X. Sun.

**Investigation:** Yufei Wu.

**Methodology:** Yufei Wu, Sean X. Sun.

**Project administration:** Sean X. Sun.

**Supervision:** Sean X. Sun.

**Visualization:** Sean X. Sun.

**Writing – original draft:** Yufei Wu.

**Writing – review & editing:** Yufei Wu, Sean X. Sun.

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
