## [Decision Letter · Decision Letter 0]

3 Dec 2024

PCOMPBIOL-D-24-00838Cell-Driven Fluid Dynamics: A Model of Active Systemic CirculationPLOS Computational Biology Dear Dr. Sun, Thank you for submitting your manuscript to PLOS Computational Biology. After careful consideration, we feel that it has merit but does not fully meet PLOS Computational Biology's publication criteria as it currently stands. Therefore, we invite you to submit a revised version of the manuscript that addresses the points raised during the review process. Please submit your revised manuscript within 60 days Feb 02 2025 11:59PM. If you will need more time than this to complete your revisions, please reply to this message or contact the journal office at ploscompbiol@plos.org. Please include the following items when submitting your revised manuscript: * A rebuttal letter that responds to each point raised by the editor and reviewer(s). You should upload this letter as a separate file labeled 'Response to Reviewers'. This file does not need to include responses to formatting updates and technical items listed in the 'Journal Requirements' section below.* A marked-up copy of your manuscript that highlights changes made to the original version. You should upload this as a separate file labeled 'Revised Manuscript with Track Changes'.* An unmarked version of your revised paper without tracked changes. You should upload this as a separate file labeled 'Manuscript'. If you would like to make changes to your financial disclosure, competing interests statement, or data availability statement, please make these updates within the submission form at the time of resubmission. Guidelines for resubmitting your figure files are available below the reviewer comments at the end of this letter. We look forward to receiving your revised manuscript. Kind regards,William CannonAcademic EditorPLOS Computational Biology Daniel BeardSection EditorPLOS Computational Biology Feilim Mac GabhannEditor-in-ChiefPLOS Computational Biology Jason PapinEditor-in-ChiefPLOS Computational Biology  **Journal Requirements:**

At this stage, the following Authors/Authors require contributions: Yufei Wu, Morgan Benson, and Sean X. Sun. Please ensure that the full contributions of each author are acknowledged in the "Add/Edit/Remove Authors" section of our submission form.

4) Your manuscript is missing the following sections: Abstract, Introduction, Results, and Methods.  Please ensure all required sections are present and in the correct order. Make sure section heading levels are clearly indicated in the manuscript text, and limit sub-sections to 3 heading levels. An outline of the required sections can be consulted in our submission guidelines here:

5) Please upload all main figures as separate Figure files in .tif or .eps format. For more information about how to convert and format your figure files please see our guidelines: 

6) We have noticed that you have uploaded Supporting Information files, but you have not included a list of legends. Please add a full list of legends for your Supporting Information files after the references list.

7) When completing the data availability statement of the submission form, you indicated that you will make your data available on acceptance. We strongly recommend all authors decide on a data sharing plan before acceptance, as the process can be lengthy and hold up publication timelines. Please note that, though access restrictions are acceptable now, your entire data will need to be made freely accessible if your manuscript is accepted for publication. This policy applies to all data except where public deposition would breach compliance with the protocol approved by your research ethics board. If you are unable to adhere to our open data policy, please kindly revise your statement to explain your reasoning and we will seek the editor's input on an exemption. Please be assured that, once you have provided your new statement, the assessment of your exemption will not hold up the peer review process.

**Reviewers' comments:**Reviewer's Responses to Questions

Reviewer #1: In this study the authors developed a cellular active filtration mathematical model and coupled it to a simplified lumped parameter model of the circulation. Simulation results show that changes in filtration has effects on bulk blood flow.

The filtration model contains a large number of parameters. Since little experimental data currently exists to fully define all of the parameters in the detailed filtration model, the simulation results may be meaningful only qualitatively at the current time, and it is difficult to validate the construction of this model.

When coupling to the lumped parameter model, there are only a few interface parameters. This presents an opportunity to practically conduct empirical parameter tuning and validation.

The qualitative simulation results presented in this study make intuitive sense. There are large degrees of freedom in parameter tuning to obtain desired results. Since there is no validation data for the detailed filtration model, it is difficult to make definitive conclusions regarding the performance of the model presented.

Reviewer #2: Summary:

The authors have developed a mathematical model to evaluate fluid flow and hydrostatic and osmotic pressures within a network of ion and fluid pumps. The authors argue that such a network exists in the body insinuating that various epithelial and endothelial elements act as pumping units and their individual performance at the cellular scale gives rise to the steady state flow and pressure variables at the organ scale. Using a simplified circuit, they show that optimal working of the pumps is achieved at the intersection of individual pump performance curves and the system curve of the network.

Comments:

Minor comments: The plots are highly congested, and legends have very small font size, which makes it hard to read and absorb the paper.

Major comments:

1. While the abstract states that the proposed "model generates pressure and osmolarity gradients across physiological compartment”, this in my view is over-representation the model and the findings. The model assumed only two organ systems—heart and the kidney- vascular interface. The conclusions need to be toned down a bit. Furthermore, it must be emphasized that so far there is no evidence that endothelial cells can create hydraulic pressure gradients like epithelial monolayers. Therefore, the addition of endothelial pump in the circuit is rather a conjecture. The references 28 and 29 indicate a ‘pump and leak’ theory in the cornea where the high pressure causes fluid leakage from the lumen rather than active regulation.

2. The result in Fig5d don’t make sense when experimental observations are taken into consideration. Previous work on osmolarity measurements across the renal epithelial tubules shows that water transport happens isosmotically from the tubules into the blood vessels. Here, the calculations give rise to an increasing osmolarity from node 2-4. Can the authors explain this?

3. Assuming the energy input to a pump in biological terms would mean rate of ATP hydrolysis, can the model predict dynamics of optimal energy consumption and the limits around stall pressure? What would that mean in a physiological context?

Reviewer #3: The manuscript "Cell-Driven Fluid Dynamics: A Model of Active Systemic Circulation" presents a model of fluid and salt transport across an epithelium, intending to use this tissue-scale model as the basis for simulating fluid circulation in animals with organs like the kidney and intestine. The authors argue that this work is novel in its investigation of an uncharted interaction between osmolarity and pressure within a circulatory system. However, after a thorough review, I find that the manuscript is not yet suitable for publication.

My primary concerns are as follows:

1. Lack of Novelty and Insufficient Literature Review: Fluid transport across epithelial layers is a well-established topic in physiology, with extensive foundational research. The manuscript does not cite crucial literature, making it unclear whether any genuinely novel findings are presented. The coupling between ion transport and pressure has been explored previously in equilibrium contexts. For example, [Marbach and Bocquet (2019)] offer a comprehensive review of osmotic processes and their coupling with pressure, which has already been applied to epithelial contexts (e.g., [Larsen et al., 2000]). If this study’s approach is indeed distinct from existing works, the authors must clarify these similarities and distinctions, specifically addressing how their approach advances current knowledge.

2. Misattributions and Overlooked Foundational Work: The authors reference Ref. 15 in the manuscript to justify the equation J_w on page 6, which indicates a lack of familiarity with established literature on fluid dynamics across cellular boundaries. The work by [Kay and Blaustein (2019)] offers a historical perspective on the development of this equation in the context of cell volume regulation, and [Fischbarg (2010)] provides a review of fluid transport across epithelial layers. Citing these sources, among others, would help ground the manuscript in the existing body of knowledge.

3. Ion transport mechanisms within the kidney and intestine have been extensively modeled, with advanced frameworks that incorporate feedback and regulatory processes. For example, the modeling approach described by [Marbach and Bocquet (2016)] on kidney filtration includes a literature review that could serve as a valuable resource. The authors should consult recent studies in this field to better position their model and to highlight any novel contributions, which currently remain ambiguous.

4. The manuscript suffers from poor organization and frequent grammatical errors, which hinder comprehension. Significant revisions are necessary to clarify the logical structure and to enhance readability.

5. To strengthen the manuscript, the authors should distinguish between established experimental inputs and hypothetical assumptions. For instance, the observation cited in Ref. 3, suggesting pump reorganization with increased pressure, is intriguing. The manuscript could benefit from focusing on this specific experimental insight or, alternatively, providing a broader justification for the validity of this observation. If this assumption is speculative, the authors might analyze a simplified, toy model to determine the conditions under which such assumptions would hold relevance for real-world systems.

In summary, this manuscript requires substantial revision to be considered for publication. The authors should comprehensively review relevant literature, improve the clarity of their model’s novelty, address both structural and grammatical weaknesses, and present a more coherent justification of their findings relative to existing knowledge.

References:

1. Marbach, S., & Bocquet, L. (2019). Osmosis, from molecular insights to large-scale applications. Chemical Society Reviews, 48(11), 3102-3144]

2. arsen, E. H., Sørensen, J. B., & Sørensen, J. N. (2000). A mathematical model of solute coupled water transport in toad intestine incorporating recirculation of the actively transported solute. The Journal of General Physiology, 116(2), 101-124

3. Kay, A. R., & Blaustein, M. P. (2019). Evolution of our understanding of cell volume regulation by the pump-leak mechanism. Journal of General Physiology, 151(4), 407-416.

4. Fischbarg, Jorge. "Fluid transport across leaky epithelia: central role of the tight junction and supporting role of aquaporins." Physiological reviews 90.4 (2010): 1271-1290.

5. Marbach, S., & Bocquet, L. (2016). Active osmotic exchanger for efficient nanofiltration inspired by the kidney. Physical Review X, 6(3), 031008.

**Have the authors made all data and (if applicable) computational code underlying the findings in their manuscript fully available?**

Reviewer #1: **No: **

Reviewer #2: Yes

Reviewer #3: None

PLOS authors have the option to publish the peer review history of their article (what does this mean?). If published, this will include your full peer review and any attached files.

Reviewer #1: No

Reviewer #2: No

Reviewer #3: No

**Figure resubmission:** While revising your submission, please upload your figure files to the Preflight Analysis and Conversion Engine (PACE) digital diagnostic tool, https://pacev2.apexcovantage.com/. PACE helps ensure that figures meet PLOS requirements. To use PACE, you must first register as a user. Registration is free. Then, login and navigate to the UPLOAD tab, where you will find detailed instructions on how to use the tool. If you encounter any issues or have any questions when using PACE, please email PLOS at figures@plos.org. Please note that Supporting Information files do not need this step. If there are other versions of figure files still present in your submission file inventory at resubmission, please replace them with the PACE-processed versions.
---

## [Decision Letter · Decision Letter 1]

6 Mar 2025

Dear Dr. Sun,

We are pleased to inform you that your manuscript 'Cell-Driven Fluid Dynamics: A Model of Active Systemic Circulation' has been provisionally accepted for publication in PLOS Computational Biology.

Best regards,

William Cannon

Academic Editor

PLOS Computational Biology

Daniel Beard

Section Editor

PLOS Computational Biology

Reviewer's Responses to Questions

**Comments to the Authors:**

Reviewer #2: The authors have addressed my concerns and sufficiently improved the manuscript for publication.

**Have the authors made all data and (if applicable) computational code underlying the findings in their manuscript fully available?**

Reviewer #2: Yes

PLOS authors have the option to publish the peer review history of their article (what does this mean?). If published, this will include your full peer review and any attached files.

Reviewer #2: No

---

## [Editor Report · Acceptance letter]

PCOMPBIOL-D-24-00838R1

Fluid and Solute Transport by Cells and a Model of Systemic Circulation

Dear Dr Sun,

I am pleased to inform you that your manuscript has been formally accepted for publication in PLOS Computational Biology. Your manuscript is now with our production department and you will be notified of the publication date in due course.

With kind regards,

Anita Estes
